# Development of Protein- and Fiber-Enriched, Sugar-Free Lentil Cookies: Impact of Whey Protein, Inulin, and Xylitol on Physical, Textural, and Sensory Characteristics

**DOI:** 10.3390/foods11233819

**Published:** 2022-11-26

**Authors:** Lívia Hajas, Csilla Benedek, Éva Csajbókné Csobod, Réka Juhász

**Affiliations:** Department of Dietetics and Nutrition Science, Faculty of Health Sciences, Semmelweis University, 1088 Budapest, Hungary

**Keywords:** green/red lentil, cookies, gluten-free, sugar-free, whey protein, inulin, xylitol

## Abstract

Gluten-free (GF) diets often become nutritionally imbalanced, being low in proteins and fibers and high in sugars. Preparing GF foods with improved nutritional value is therefore a key challenge. This study investigates the impact of different combinations of whey protein (11.9%), inulin (6.0%) as dietary fiber, and xylitol (27.9%) as a sweetener used in the enrichment of green- and red-lentil-based gluten-free cookies. The cookies were characterized in terms of baking loss, geometric parameters, color, texture, and sensory profile. The results showed that these functional ingredients had different impacts on the lentil cookies made of different (green/red) lentils, especially regarding the effect of fiber and xylitol on the volume (green lentil cookies enriched with fiber: 16.5 cm^3^, sweetened with xylitol: 10.9 cm^3^ vs. 21.2 cm^3^ for control; red lentil cookies enriched with fiber: 21.9 cm^3^, sweetened with xylitol: 21.1 cm^3^ vs. 21.8 cm^3^ for control) and color (e.g., b* for green lentil cookies enriched with fiber: 13.13, sweetened with xylitol: 8.15 vs. 16.24 for control; b* for red lentil cookies enriched with fiber: 26.09, sweetened with xylitol: 32.29 vs. 28.17 for control). Regarding the textural attributes, the same tendencies were observed for both lentil products, i.e., softer cookies were obtained upon xylitol and whey protein addition, while hardness increased upon inulin enrichment. Stickiness was differently influenced by the functional ingredients in the case of green and red lentil cookies, but all the xylitol-containing cookies were less crumbly than the controls. The interactions of the functional ingredients were revealed in terms of all the properties investigated. Sensory analysis showed that the addition of whey protein resulted in less intensive “lentil” and “baked” aromas (mostly for red lentil cookies), and replacement of sugar by xylitol resulted in crumblier and less hard and crunchier products. The application of different functional ingredients in the enrichment of lentil-based gluten-free cookies revealed several interactions. These findings could serve as a starting point for future research and development of functional GF products.

## 1. Introduction

In genetically predisposed individuals, gluten—a protein fraction present in wheat, barley, and rye—can trigger several disorders, including celiac disease. Certain epitopes in these proteins can initiate an immune response. Following a strictly gluten-free-diet can help control both intestinal (e.g., abdominal pain, bloating, chronic diarrhea) and extraintestinal symptoms (e.g., weight loss, delayed growth, iron deficiency anemia, weakness, muscle cramps, infertility, thyroiditis) [1]. The prevalence of biopsy-proven celiac disease has been estimated to be 0.77% worldwide, strongly varying from one region to another (e.g., 0.05% in Japan and 2.56% in Sweden) [2]. The number of people diagnosed with food allergies and intolerances is increasing day by day, and even more people have a tendency to focus on a healthy lifestyle. Customers tend to make their choices not only based on prices or preferences, but also on the composition of the product. At the same time, “free from”, including gluten-free (GF), products are consumed not only for health reasons, but as part of a “health-conscious” diet culture. While a gluten-free diet is the cornerstone of therapy for celiac patients, up to 30% of Americans report having attempted to eliminate or reduce the amount of dietary gluten without any medical indication. However, most GF products are not “healthy”: they are usually rich in calories and poor in many nutrients [3]. As they are mainly based on rice and corn flour and starch, they have a low fiber content [4]. Their added sugar and saturated fat content, on the other hand, is usually too high. Due to their advantageous physiological and sensory characteristics and good technological qualities as thickeners or humectants, their resistance to heat and pH fluctuations, sugar alcohols—among which xylitol occupies an important position—are extensively employed by the food industry. Their success can also be attributed to consumers’ growing distaste for artificial additives, including intense sweeteners [5]. Xylitol has been recently attracting a growing interest and seems to be a proper choice for sugar substitution in gluten-free products as well [6,7].

The absence of gluten also results in a lower protein content. If gluten sensitivity is accompanied by a carbohydrate metabolism disorder, there is still a reduced range of ready-made, so-called convenience biscuit products. On the other hand, bread consumption among celiacs is significantly lower compared to the healthy population. In contrast, both sweet and savory biscuits’ consumption was shown to be significantly higher in the celiac group, this revealing that gluten-sensitive patients mostly cover their carbohydrate intake with biscuit-like products [8]. From 2020 to 2027, the gluten-free (GF) market is anticipated to expand at a compound yearly growth rate of 9.2% [4]. The global gluten-free products market size was valued at USD 5.9 billion in 2021 and is estimated to reach USD 7.5 billion by 2027 [9,10]. Although the choice of such foods is in continuous development, it is still a challenge to ensure that a GF product is high in protein and fiber and low in sugar. Different protein and fiber enrichments affect both physicochemical and sensory properties. Fiber-enrichment in GF cakes containing inulin resulted in improved specific volume, harder crumb, and brighter crust [11]. However, other authors found lower specific volumes in inulin-enriched breads, but on the other hand, higher fiber addition led to a physical and/or chemical impediment to digestion [12]. Proteins from animal sources such as whey and casein provide good technofunctional qualities in breads, such as solubility, gelling, and water-holding ability. On the other hand, bread blends with legume proteins were shown to have higher water absorption, weakening the dough strength and stability, thus altering the bread texture properties. One of the biggest obstacles to improving the protein content of bread is the potential alteration of specific volume, texture profile, color, or sensory attributes [13].

More nutrient-dense flours could be employed in GF bakeries to improve their nutritional profile. Di Cairano et al. report on GF biscuits created with different underutilized flours using buckwheat, sorghum, millet, lentil, and chickpea flours. Their experiments showed a good acceptance of these GF products, among which lentil-containing biscuits were the most preferred [4]. As potential carbohydrate sources, lentils (*Lens culinaris* Medik.) excel in terms of protein, fiber, folate, zinc, and iron content [14], as well as bioactives, such as polyphenols [15,16]. Nevertheless, in spite of their high nutritional value and low ecological footprint [17], lentils are not commonly used in bakery products, which may be due to different reasons such as the unusual taste, higher price compared to other pulses, or presence of antinutritive compounds (i.e., protease inhibitors) [18,19,20].

The above-mentioned Italian research group reports that legumes can be used to create GF biscuits that are acceptable and that may also improve their nutritional value, antioxidant activity, and glycemic index [18]. Another study examines the use of brown lentil flour in mixtures with wheat flour that contain up to 25% lentil flour. In comparison to the wheat product, the resulting cookies were harder and had different color features [21]. Our previous results show that GF biscuits can be successfully produced from different lentil flours (green, red, black, yellow, and brown) without adding any other flour. All the lentil flours were superior to the rice control in several terms, including sensory acceptability for some of the lentils [22].

The aim of this research was to further enhance the nutritionally favorable properties of lentil-based cookies in terms of increasing their protein (by adding 11.9% whey protein isolate) and fiber content (by adding 6.0% inulin fiber), as well as to develop a sugar-free product (by replacing sugar entirely by xylitol). Our study also aimed to gain a deeper insight into the functionality of the different ingredients and their interactions. The impact of protein (whey), fiber (inulin), sweetener (xylitol), and their combinations on the physical (geometry, baking loss, color, texture) and sensory attributes of lentil-based cookies were evaluated.

## 2. Materials and Methods

### 2.1. Materials

Lentils were purchased at a local market in Budapest, Hungary. The country of origin was Canada. The lentil seeds were milled and sieved with a 500 μm particle size mesh to produce flour as previously described [22]. Xylitol, powdered sugar, glucose, margarine, salt, and sodium bicarbonate were purchased from local supermarkets. Inulin (Orafti^®^FTX, long chain, inulin content: 98%, sweetness level: 0%) was provided by Beneo (Tienen, Belgium). Whey protein isolate (protein content: 91.7%) was purchased from a body building webshop (wheyprotein.hu accessed on 15 February 2020)

### 2.2. Preparation of Cookies

Lentil-based cookies were prepared as previously reported [22]. Control cookies were enriched with fiber and protein according to the following:

There was 12.6 g inulin incorporated into 87.4 g (instead of 100 g) of flour, replacing thus part of the lentil flour. This inulin level was chosen in line with the relevant EU regulation, corresponding to a ratio of 6% (*w*/*w*) of the total ingredients, which is the criterion for qualification as a product “high in fiber” [23].

There was 25.0 g whey protein isolate incorporated into 75.0 g (instead of 100 g) of flour. This amount was chosen in order to qualify the product as a “source of protein” [23].

For sugar-free cookies, sugars (saccharose and glucose) were entirely replaced by the corresponding amount of xylitol, taking into account their equivalent sweetening power [24].

The cookies were prepared according to 8 recipes, as summarized in Table 1. The dough was laminated to a 6 mm thickness and cut in circles of 50 mm of diameter with a cookie cutter. Twelve pieces were obtained from each sample type. They were baked for 10 min at 200 °C (static) in an electrically heated oven.

Cookies were placed in plastic bags and stored at room temperature. All investigations were performed on the following day.

### 2.3. Methods

#### 2.3.1. Color Measurement

The surface color of each individual cookie was measured in the CIELab color space using the Chroma Meter CR-410 (Konica Minolta, Inc., Tokyo, Japan). The instrument was calibrated to a white standard plate (CR-A44) before measurements. L* is a measure of the lightness from black (0) to white (100), while a* describes the redness (positive) or greenness (negative) and b* describes yellowness (positive) or blueness (negative). The color differences perceived by the human eye between two sample types, Delta E (∆E), was calculated using the mean value of L*, a*, and b* obtained for the twelve replicates (baked cookies). There is no color difference perceived by the human eye if ∆E < 1, while a color difference is clearly visible if ∆E > 3 [25].

#### 2.3.2. Determination of Baking Loss and Geometry

These were determined according to the previously described method [22]. Baking loss was characterized as the cookie weight reduction after baking. The diameter was obtained by taking two measurements from one cookie in 90° rotations. The height (thickness) was recorded at the center point and the edges. Twelve cookies were measured from each sample type. The spread ratio of a cookie was calculated by dividing the mean of two diameter values by the mean of three height values.

#### 2.3.3. Texture Analysis

Texture profile analysis was carried out by using the Brookfield CT3 Texture Analyzer (Ametek Brookfield, Middleborough, MA, USA) equipped with a TA9 probe (stainless steel needle, diameter: 1.5 mm). Two cycles of 5 mm penetration were performed. The trigger load for the probe was 4 g. The test speed was set at 1 mm/s. Four cookies per sample type were randomly selected, and the measurement was replicated four times in the central position of each cookie. The data were recorded and analyzed by using the TexturePro CT v1.9 build 35 software (Ametek Brookfield, Middleborough, MA, USA) to calculate certain texture parameters.

The texture profile of the samples was determined by recording the load (g) as a function of time (s). The texture parameters were defined by the software and calculated as described in our previous study [22]. Briefly, hardness (g) is the maximum force observed in the first mastication cycle; adhesive force (g) is the peak of negative force measured in the first mastication cycle; cohesiveness (-) is the ratio of undercurve areas of positive force during the second to that during the first compression.

#### 2.3.4. Sensory Tests

Sensory analyses were performed from freshly baked cookies, distributing the samples from the same baking lot between panel members after its homogenization. Samples were allocated 3-digit random sample codes. A simplified profile analysis was applied during the sessions, i.e., the sensory descriptors (required by ISO 11035) were previously established and the panelists were given a complete list of the attributes [26]. The panelists (13 for green lentil cookies and 16 for red lentil cookies) consisted of students and staff members of the Department of Dietetics and Nutrition Science. The sensory tests were carried out meeting the criteria of the relevant ISO standards [27]. The sample characteristics were compared to internal reference values determined previously for the reference sample (sugar-containing control cookie containing no added inulin or protein). Neutral mineral water was used as a taste neutralizer between evaluations. The values of sensory attributes were evaluated on a 1–100 structured linear scale (values were given in multiples of 10). The sensory descriptors used on the scoresheet were as follows: surface homogeneity: 1 = cracked surface, 100 = homogeneous, smooth surface; surface color: 1 = greenish-yellowish, 100 = reddish brown/brown; baked aroma: 1 = uncharacteristic, not perceptible baked aroma, 100 = intense roasted aroma; lentil aroma: 1 = uncharacteristic, not perceptible lentil aroma, 100 = intense lentil aroma; lentil taste: 1 = uncharacteristic, not perceptible lentil taste, 100 = intense lentil taste; sweet taste: 1 = uncharacteristic, not perceptible sweet taste, 100 = intense sweet taste; hardness: 1 = soft, 100 = hard; crunchiness: 1 = sticky, chewy, 100 = crunchy; crumbliness: 1 = crumbly, dry, 100 = not crumbly. The panelists were asked to rank samples according to the decrease of overall liking, so that the most appealing sample received the lowest score. At the end of the scoresheet, an open-ended textbox was provided to describe any additional perceptions and attributes.

### 2.4. Data Analysis

In the case of geometry parameters, the two (diameter) or three (height) measured data of one cookie were averaged. These mean values were used for statistical analysis (12 data per sample type).

In the case of texture parameters, the data interquartile range was used as the screening method prior to further analysis. Sixteen measured data were arranged in increasing order, and the lowest and the highest two data were skipped during analysis.

Statistical analysis of the data was performed with Statistica ver.13.5.0.17. (TIBCO Software Inc., Palo Alto, CA, USA) software. In the present experiment, there were three factors (xylitol, inulin, whey protein), and each factor had two levels (included or not included). In the case of physical and textural parameters, factorial analysis of variance (ANOVA) was used to examine the interactions among experimental factors (cookie ingredients). The partial eta-squared (η^2^_p_) from the factorial ANOVA was used to quantify the effect size. The magnitude of the effect sizes was interpreted as follows: small = 0.01, medium/moderate = 0.09, large = 0.25 [28]. Sensory data were analyzed by the one-sample Student’s *t*-test to compare different sample types with the reference sample (control). Tukey’s honestly significant difference (HSD) post hoc test was applied to determine the presence of statistically significant differences between sample types. The data from the preference ranking test were analyzed using Friedman’s test. The significance level (α) of 0.05 was set for all analyses.

## 3. Results and Discussion

### 3.1. Baking Loss and Geometry

The enrichment of lentil cookies by the combination of protein, fiber, and sweetener resulted in minimal differences in baking loss and geometry compared to the control.

For green lentil cookies baking loss varied between 16.3 and 18.2% compared to 17.0% for the control (see Table 2). There were no significant changes whether sugar was replaced by xylitol (16.9%) or whey protein isolate (WP) was added (16.9%) to the control; however, inulin enrichment led to significantly higher baking loss (18.2%). Water loss was not found to be significantly different in wheat cakes or biscuits where sugar, respectively wheat flour, was partly replaced by inulins [29,30]. However, Laguna et al. found a significant increase of water loss during baking for wheat-based biscuits enriched with different fibers (wheat and apple fibers) already at the 5% level. They attributed the water loss to the water retention capacity of fiber [31]. This water retention capacity of the inulin used in our experiments may be lower than the water retention capacity of the lentil flour, which was partly replaced by inulin. Stuck and co-workers also observed higher baking losses during fiber enrichments with different fibers [32]. Ho and Abdul Latif found variable baking losses in cookies with increasing fiber content and considered lower moisture content as favorable for cookies from the point of view of more shelf-stable products [33]. Water loss also increased in gluten-free cookies enriched with fruit pomaces as fiber sources [34]. Regarding cookies enriched with both fiber and protein, the presence of WP seems to compensate for the effect of inulin (baking loss for GL-IN-WP: 16.6%), with xylitol showing the same effect (baking loss for GL-XY-IN: 17.4%). Maravić et al. also observed an increase of moisture content upon the addition of whey protein [35]. Compared to the control, no significant baking loss changes were observed in any of these samples; however, a significantly lower value was obtained for the double-enriched and sugar-free product (GL-XY-IN-WP: 16.3%)

The diameters of green-lentil-based cookies varied between 48.7 and 64.3 mm, and the control diameter was 61.5 mm (Table 2). The diameter decreased significantly when xylitol was used instead of sugar (48.7 mm), which is in line with what was observed by Zoulias and co-workers, who explained the phenomenon by the effect of xylitol on the setting time of the dough [36]. Inulin addition increased spreading significantly for both sugar- and xylitol-containing cookies (diameters: 64.3 mm and 58.5 mm, respectively). Blanco Canalis and co-workers found a similar diameter increase when 12% of wheat flour was replaced by inulin (Orafti HP). They assumed that the solubility of inulin, as a soluble fiber, increases with temperature; thus, it is progressively dissolved during baking, this resulting in a rise of the spread rate (resulting in lower height and higher diameter values), improving biscuit quality [37]. The addition of WP did not exert any significant changes on the diameter; however, the diameter decreased significantly in xylitol-containing cookies enriched with WP as a result of the effect of the xylitol (62.8 mm for GL-WP vs. 58.0 mm for GL-XY-WP and 55.5 mm for GL-XY-IN-WP). The neutral effect of WP on the diameter is in line with results obtained for wheat-based cookies where the presence of 5–15% of whey protein did not cause any changes in diameters [38]. Cookies containing combined functional ingredients had overall smaller diameters (55.5–58.5 mm) compared to the control, except the sugar-based one, in which the simultaneous presence of inulin, WP and xylitol induced an increase in diameter (62.4 mm).

The heights of green-lentil-based cookies (Table 2) changed between 5.1 and 8.3 mm (7.1 mm for the control). Height was significantly reduced (5.9 mm) when sugar is replaced by xylitol. Struck and co-workers explained the height decrease upon sugar replacement by alternative sweeteners by earlier structure setting during baking, presumably caused by a lower starch gelatinization temperature [32]. Gong et al. also associated for this phenomenon with thermal gelatinization temperatures: lower gelatinization temperatures were observed for xylitol, which inhibits the formation of higher cookies [39]. Compared to the control, the cookies were significantly higher in the presence of WP only in sugar-containing (8.3 mm) samples or in samples with both WP and inulin (8.1 mm for GL-IN-WP, 8.2 mm for GL-XY-IN-WP). WP also induced an increase in height in xylitol-containing samples (5.9 mm for GL-XY vs. 7.7 mm in GL-XY-WP), partly counterbalancing the effect of xylitol. This effect of WP on the height of cookies is not in line with the observations of Komeroski and co-workers, who did not find any significant differences upon adding 10 or 20% WP to bread compositions based on chickpea–cassava flour or to wheat flour [40]. However, the literature reports are contradictory, as Ahmed et al. and Maravić et al. registered lower height values in wheat cookies enriched with 10–15%, respectively 20% WP; however, the difference may also be attributed to the absence of gluten in our case [35,38]. As a possible effect of WP, Maravić referred to the hydrogen bonds formed between WP and water, which may lead to a protein network that could modify the spreading properties in cookies. As the gluten network was not present in our products, this may lead to an inverted effect produced by WP, i.e., an increase in height. Inulin led to a significant decrease in height only in sugar-containing products (5.1 mm). Regarding the effect of inulin, our observations are in line with those obtained by Tsatsaragkou et al., who replaced part of the sugar by inulin and noticed a decrease in cookie height [29]. Blanco Canalis et al. obtained a similar height decrease (and diameter increase) upon the addition of inulin to wheat-based cookies [30,37], as well as Handa et al. [41]. However, the presence of xylitol compensated for this effect of inulin (6.5 mm for GL-XY-IN). The same, but even stronger compensating effect applies for WP (GL-IN-WP, GL-XY-IN-WP).

Cookie volumes varied between 10.9 cm^3^ and 25.8 cm^3^ (21.2 cm^3^ for the control). The production of sugar-free cookies resulted in a volume reduced by almost half (10.9 cm^3^) of the original. This can be attributed to the function of sucrose in baked goods, acting not only as a sweetener, but also controlling the gelatinization of starch, thus influencing the physical properties, especially the volume. In the presence of sucrose, starch gelatinization is a longer process, allowing thus a higher vapor pressure from water vapor and carbon dioxide during baking. As a result, air bubbles can grow in sugar-sweetened cakes or cookies, enabling the development of a voluminous cake structure [29]. Starch gelatinization is also crucial in gluten-free formulations, where the addition of certain types of starch in the gluten-free formula could improve the batter consistency [42]. Struck et al. obtained similar results with sugar-reduced cakes, concluding that substitution of sucrose weakens the stability of air bubbles and leads to an earlier rupture of the cell walls; thus, the gas-holding ability during baking is reduced [32]. Kweon and Levine also emphasized the positive role of sugar in starch gelatinization during the texture development of cookies [43]. The volume was also decreased by the enrichment with inulin for sugar-containing cookies (16.5 cm^3^); this effect was slightly compensated by xylitol (17.4 cm^3^ for GL-XY-IN). Similar to the impact on height, WP contributed to a significant volume increase, the highest volume being reached for GL-WP (25.8 cm^3^) and GL-IN-WP (24.8 cm^3^) cookies.

The spread ratio or “biscuit factor” (diameter/height) was determined to show shape changes during baking compared to the original value for raw dough (8.3 for our cookies). It is used to determine the quality of flour: higher spread ratios are desirable for cookies [44]. The spread ratio varies between 6.9 and 12.8 for green lentil cookies, and the control spread ratio was 8.6. Inulin produced a significant increase of spreading (12.8), while WP induced a significantly lower ratio (7.6). In the combined cookie, the simultaneous presence of xylitol and WP resulted in the lowest ratio (6.9). The results confirmed the above-mentioned observations: the addition of inulin increased the diameter, but not the height of the cookies, thus increasing the spread ratio, while WP did not influence the diameter, but increased the height, thus decreasing the spread ratio.

Regarding red-lentil-based cookies, a similar range was obtained for the baking loss values (16.9–17.9%; 16.9% for the control, see Table 3), while the presence of xylitol resulted in no significant change. As stated by other authors as well, xylitol-dominated blends produced doughs very similar to the sucrose-containing control [45]. This is partly explained by others as the influence of solubility on batter stability, xylitol having a solubility close to that of sucrose [46]. Enrichment with either WP or inulin produced higher baking losses (17.8% and 17.4%, respectively). By applying combined enrichments, overall higher baking losses were obtained in the presence of xylitol; however, the baking loss did not differ from the control in sugar-containing double-enriched cookies (baking loss for RL-IN-WP: 17.0%).

The diameters of red-lentil-based cookies varied between 58.0 and 67.4 mm, and the control diameter was 63.2 mm (Table 3). The presence of xylitol led to a decrease in diameter (58.0 mm), similar to green-lentil-based cookies. Diameters increased in the presence of inulin, whether used with sugar (67.4 mm) or xylitol (66.8 mm). WP enrichment did not contribute to any changes in diameter, whether in combinations or not.

The heights of red-lentil-based cookies changed between 6.1 and 8.4 mm, and the control height was 6.9 mm (Table 3). Xylitol had a different impact from that observed for green-lentil-based cookies, i.e., a significant increase of height (8.0 mm). The presence of WP resulted in a significant height increase in most of the cookies (7.6 mm for RL-WP, 8.4 mm for RL-XY-WP, 7.8 mm for RL-XY-IN-WP), except from the one in which it was combined only with inulin (7.5 mm). Similar to the case of green-lentil-based products, inulin produced a decrease in cookie height, compensated by the other two added ingredients.

Red lentil cookies had volumes between 21.1 cm^3^ and 25.3 cm^3^ (21.8 cm^3^ for the control). Interestingly, there were no significant differences observed by the addition of either inulin or xylitol; the only significantly different volumes were found for cookies with WP (RL-WP: 25.0 cm^3^, RL-XY-WP: 25.0 cm^3^, RL-IN-WP: 25.3 cm^3^). WP exerted a similar volume-increasing effect as observed for green lentil cookies.

Regarding spread ratio, similar tendencies were observed as for green lentil cookies: inulin produced an increase (11.1), while WP led to a decrease of the ratio (8.6); however, the latter was not significant. In red lentil cookies, xylitol also exerted a stronger decreasing effect on this parameter, producing a significantly lower biscuit factor on its own (7.3) and in combinations (RL-XY-IN: 10.5, RL-XY-WP: 7.4, RL-XY-IN-WP: 8.0). The spread ratios for red lentil cookies varied within limits similar to the green ones (7.3 and 11.1), and the control spread ratio was somewhat higher (9.2).

The interactions occurring when different enrichments were applied were analyzed based on the partial eta-squared results of the factorial ANOVA (see Figure 1; the interaction plots for the data of Table 2 and Table 3 can be seen in Appendix A). The interactions between different ingredients used for the enrichment of cookies are rarely analyzed, as in most cases, only one compound is changed in recipes. However, Gómez and Sciarini reported that the quality of gluten-free products is influenced by the interactions between ingredients (i.e., starch, sugar, proteins, etc.), and therefore, the effect of certain compounds should be evaluated in each product individually [47].

Analyzing our data, it can be concluded that the direction of the changes is the same for both lentil types. WP contributes to maintaining lower baking loss when used with inulin, i.e., there are no statistically significant differences between the control and IN-WP combinations (effect size of IN×WP interaction: 0.35 for green/large/and 0.20/moderate/for red lentils). If xylitol is used as a sweetener, there is no statistically significant difference between the baking loss of enriched cookies and cookies containing only xylitol, irrelevant of the type of lentil used. For red lentil cookies, sugar- and xylitol-containing samples showed the same trend. Therefore, only this INxWP interaction is significant here. However, the XYxIN and XYxWP interactions are only significant for green lentil samples, where in the presence of xylitol, both WP and inulin produce a different effect. In the presence of xylitol, inulin no longer causes a significant increase, while the effect of WP becomes significant and reduces the loss. Compared to the large effect of the INxWP interaction (0.35), the effects of XY-WP (0.09) and XY-IN (0.13) are only medium.

Xylitol induces a decrease of the diameter; however, this effect is inverted when it is used in combinations. The INxWP interaction is significant for both lentil types (0.46 and 0.24 for green and red lentils, respectively). The diameter-increasing effect of inulin is counteracted in the presence of WP. The XYxIN and XYxWP interactions are only significant for the green lentil cookies, and their effect is much smaller (0.08 and 0.14 for green and red lentils, respectively) than observed for the INxWP interaction.

The cookie height increased by WP, but this effect was counteracted by the interaction with inulin. The INxWP and XYxIN interactions are significant, but only the latter and only for the green lentil cookies is large (0.36). In the presence of sugar, the addition of inulin reduced the height considerably. Even though xylitol compensated for this effect, the height of the XY-IN sample was still significantly lower compared to the control. The XYxIN interaction is significant in the case of red lentil cookies as well, but its effect is small (0.08). The INxWP interaction is moderate for both lentils (0.10 and 0.14).

The cookie volumes were generally increased upon the addition of WP, either for sugar-containing or sugar-free combinations. Inulin and especially xylitol had markedly different effects on cookies made from green or red lentils, their effect having been more pronounced on the green lentil products. While none of the interactions were significant for red lentils, for green lentils, the XYxIN interaction is significant and large (0.31). In the presence of sugar, inulin addition resulted in a decrease in volume, while in the presence of xylitol, a volume increase was observed upon inulin addition. For both lentil types, the simultaneous presence of the three components (xylitol, inulin, WP) did not cause a significant change in the volume when compared to the control.

The spread ratio was positively impacted by the presence of WP (flatter cookies) and negatively influenced by inulin for both lentil types. The negative effect of the latter was compensated in combinations with WP and xylitol (especially for red lentil cookies). For both lentils, the INxWP interaction had a significant and large effect (0.37 and 0.33 for green and red lentil cookies, respectively). In the presence of WP, the effect of inulin was counterbalanced, thus no significant change was observed in comparison with the control. In the case of green lentil samples, the effect of XY-IN is also large (0.27); thus, not only WP, but also xylitol acts against the effect of inulin. The XY-IN interaction for the red lentils is also significant, but only small (0.07.)

### 3.2. Color (L*, a*, b*) Parameters

Color is a significant quality attribute because it can arouse an individual’s appetite. It is one of the process indicators during baking and roasting, as brown pigments are formed as the browning and caramelization reactions advance [48].

Positive a* and b* readings demonstrated that redness and yellowness predominated in cookie samples irrelevant of the lentil type. The Maillard reaction or non-enzymatic browning, which depends on the amount of reducing sugars and amino acids or proteins on the surface, baking temperature, and baking time, may stay in the background for the red coloring, as well as the air velocity in the oven and the ingredient composition [44,48].

In the case of sugar-containing green lentil cookies, the addition of xylitol (Table 4), WP or inulin did not induce any significant changes in L* compared to the control (35.68). The addition of the latter two did not produce a change in L* in sugar-free cookies either, and double-combined sugar-free cookies (35.58) did not differ significantly from the control. Even though Di Cairano and co-workers found higher L*, a*, and b* values for cookies enriched with soluble inulin, they concluded that there is no definite trend in the literature regarding the impact of inulin on the color of baked products [49]. The limited effect of inulin on the color is also explained by the restricted rate of the Maillard reaction in which sugars originating from inulin could serve as raw materials; on the other hand, protein–inulin interactions can also influence color [50]. Furthermore, the a* and b* values decrease upon replacement of sugar by xylitol and increase by WP addition. Inulin affected only the b* value (GL-IN: 13.13). WP addition on its own did not result in darker cookies, although other authors found that it would decrease lightness in WP-enriched corn-flour-based cookies. They attributed this effect to the high lysine content in WP: the ε-amino groups in lysine provide a source of free amino groups that react with the carbonyl groups of reducing sugars during the Maillard process [51]. However, Komeroski et al. report no changes in the color attributes of chickpea–cassava flour bread with 10–30% WP [40]. This supports our findings, leading us to the hypothesis that the Maillard reaction (leading normally to darkening and more reddish colors) has a limited role in the formation of the color of lentil-based cookies compared to cereal-based ones. This is further supported by the low hydroxymethyl-furfural values previously found by us in lentil cookies [22]. The effect of the Maillard process is more visible in the a* and b* values, which are higher, in the presence of WP, in accordance with the results of Sahagún et al. [51] and Komeroski et al. [40]. In samples with xylitol, both the a* and b* values generally decreased in most of the combinations. The color changes induced by xylitol are attributed to the absence of the melanoidins formed in the Maillard reaction of sugar-containing cookies. Clearly perceptible color differences were obtained for each of the combinations compared to the control (∆E: 3.3–9.2), except for the sugar-free, double-enriched cookies, which showed a perceivable, but not clearly visible color difference (1.9).

In the case of red lentil cookies (Table 5), the different lentil color led to generally lighter, more reddish, and yellowish cookies, as observed previously [22]. Otherwise, the trends were similar to those obtained for green lentil cookies, i.e., no significant changes upon enrichment with either inulin or WP. Here, however, the replacement of sugar by xylitol resulted in a more pronounced effect on lightness in any of the xylitol-containing combinations (L*: 48.05–53.01 compared to 45.25 for the control). Cookies were less reddish using xylitol in any combination (a*: 7.61–14.01 compared to 16.53 for the control). This effect of xylitol was somewhat attenuated by the presence of inulin or WP, especially by their combination (a*: 16.67 for RL-XY-IN-WP). On the other hand, cookies were more yellowish in the presence of xylitol (b*: 32.21–36.17 compared to 28.17 for the control). WP or inulin had no impact on the b* values when used as single enrichments, but produced higher values in combinations with xylitol. Visually perceivable differences covered a wider range (∆E: 2.7–12.5) than in the case of green lentil cookies, i.e., different functional ingredients had higher impacts on the color attributes of some red lentil cookies. However, color differences were not clearly visible for many cookies (∆E < 3). The observed differences were the highest in xylitol-containing cookies, due to considerably higher lightness and lower redness values. Thus, sugar replacement by xylitol has been shown to compromise the visual aspect of lentil-based cookies.

Regarding the interactions occurring upon applying different enrichments (see Figure 1; the interaction plots for the data of Table 4 and Table 5 can be seen in Appendix A), it can be concluded that, in terms of all color attributes, the replacement of sugar by xylitol had the highest impact (overall large effect sizes between 0.63 and 0.93, except for L* in green lentil cookies, which is moderate/0.24/). Large effect sizes exerted by XYxIN (0.64) and XYxWP (0.80) combinations were observed on the redness of red lentil cookies, a* values having been increased in sugar-free cookies upon either inulin or WP addition.

### 3.3. Textural Properties

The texture parameters were calculated from the texture profile recorded instrumentally and are summarized in Table 6 and Table 7.

Similar to those reported by many others, our green lentil cookies (Table 6) became harder by inulin enrichment [31,49]. Hardness decreased upon WP addition; however, other formulations reported that hardness increased by WP enrichment [35,51]. Cookies containing xylitol were softer (lower hardness), as many other authors found. This property is attributed to the higher affinity of xylitol for water [7,36].

Regarding stickiness, two cookies were significantly stickier (higher adhesive force) than all the others, both containing WP: GL-XY-WP and GL-XY-IN-WP. In terms of crumbliness, all the xylitol-containing cookies showed higher cohesiveness values, this indicating a less crumbly texture.

As observed for the geometrical parameters, the effects of the different functional ingredients were counteracted in combined enrichments, so that the hardness of GL-XY-IN-WP (639.4 g) did not differ significantly from that of the control (612.0 g). However, this was not reflected in the sensory test (Table 8).

Regarding the textural attributes of red lentil cookies (Table 7), the same tendencies were observed as for the green lentil products, i.e., softer cookies upon xylitol and WP addition, increased hardness upon inulin enrichment. Here, however, all the combinations resulted in significantly softer cookies compared to the control. Stickiness was also different from that observed for green lentil cookies: here, RL-XY, RL-WP, and RL-IN-WP were less sticky than the control. Finally, xylitol showed the same effect as for green lentil cookies in terms of crumbliness: xylitol-containing red lentil cookies were all less crumbly than the control.

Regarding interactions (see Figure 1; the interaction plots for the data of Table 6 and Table 7 can be seen in Appendix A), XYxWP exerted a significant and large interaction on hardness for both lentil types (0.63 and 0.42 for green and red lentil cookies, respectively). Hardness also decreased under the influence of xylitol and WP. In the case of green lentil cookies, adding WP to sugar-free samples increased hardness, but the hardness of the XY-WP-containing samples was still lower than the hardness of the control sample. In the case of red lentils, adding WP to the samples containing xylitol did not change the hardness. Even if gluten is absent, protein-starch interactions can play a significant role in texture formation. Proteins can retain water and form a gel, thus stabilizing the starch [40]. On the other hand, it was stated that the type of starch also impacts the efficiency of whey protein on several attributes, including thermo-mechanical properties, specific volume, and firmness [52]. Starch composition has been shown to be different in wheat breads enriched with green lentil flour compared to the wheat control [53]. Therefore, the combined effect of starch-protein can have a unique impact on texture, including hardness, which depends on both ingredients. In our case, the clear trend of hardness increase upon WP enrichment was confirmed for both lentil types.

### 3.4. Sensory Properties

The sensory profile analysis of green lentil cookies prepared with combined enrichments (Table 8, Appendix A) showed no significant differences in terms of baked aroma, lentil taste, and crumbliness. Differences were significant for surface homogeneity (GL-IN-WP having been the most homogeneous), surface color (GL-IN-WP was found the most reddish-brownish, this reflecting the high a* value obtained instrumentally), lentil aroma (the most intensive lentil aroma was found for GL-XY-IN, which does not seem to be in connection with perceived brownish color), sweet taste (GL-IN-WP was found to be the sweetest), hardness and crunchiness (for both, the highest ranking was obtained for GL-IN-WP). Cookies sweetened with xylitol preserved the greenish color of the lentils compared to the control; their surface was less homogeneous, and they were found to be softer, less crunchy, and less sweet. This is in agreement with the observations made by Winkelhausen et al., who also obtained softer and less crunchy cookies; however, they did not find any difference in sweetness [6]. In addition to softer texture, Mushtaq et al. also obtained crumblier cookies upon xylitol addition, in line with our observations [7]. The addition of inulin resulted in a more cracked surface (except from GL-IN-WP), as observed by others [30]. Combined with WP, inulin enrichment resulted in properties that were considerably different from the other combined products in terms of surface homogeneity and color, sweet taste, hardness, crunchiness, and crumbliness; however, the last four properties (together with lentil taste) were similar to those of the control. The higher hardness values obtained for GL-IN-WP cookies may also be attributed to the hardening effect of whey protein, which was observed by others [51,54]. Nevertheless, the hardness values obtained during instrumental texture analysis were not in line with those obtained during the sensory tests; thus, perception of hardness by the panelists was probably influenced by other attributes of the cookies (e.g., crunchiness, texture homogeneity). Increased crumbliness upon addition of WP was also observed by others in sponge cakes [35].

The sensory profile analysis of red lentil cookies (Table 9, Appendix A) showed significant differences for all the properties tested. Except for the RL-IN-WP combination, all the samples were less crumbly compared to the control, similar to the green lentil cookies. As for the green lentil cookies, the inulin–WP combination showed outstanding properties compared to the rest of the combinations: a more homogeneous surface, a more intensive reddish-brownish color, a less intensive lentil taste, a sweeter taste, a less soft and crunchier texture. Lentil notes and baked aroma were the most intensive for the RL-XY-IN combination. Xylitol-containing cookies were generally less sweet, less crumbly, and softer than the control. Baked aroma was the most intensive for the control, which was also the most homogeneous.

## 4. Conclusions

Our research revealed the potential of green- and red-lentil-based gluten-free cookies enriched with protein (whey protein) and fiber (inulin). The role of xylitol as an alternative sweetener was also tested regarding the formation of the measured properties. The technological parameters showed that baking loss is decreased by inulin, but for the green lentil cookies, this effect is counterbalanced when WP is simultaneously used. When xylitol is used as a sweetener, there is generally no major difference between the baking loss of double-enriched or control cookies. Cookies’ height was decreased and diameter was increased significantly when xylitol was used instead of sugar, while inulin and WP addition increased the spread ratio.

Regarding color, clearly perceptible differences were detected for all the enrichment combinations compared to the controls. All red-lentil-based cookies were lighter and more reddish and yellowish than the green-lentil-based ones. Sugar replacement induced a lighter color, especially for the red lentil cookies, while WP and inulin had no considerable effect on color when used alone, but produced significant color changes in combinations. The different functional ingredients had more pronounced impacts on the color properties of red lentil cookies, but in both cases, sugar replacement by xylitol was to the detriment of the visual aspect in terms of lightness and redness. In most of the cases, the interactions between xylitol, WP, and inulin in the XY-IN-WP cookies led to color attributes similar to the controls.

In terms of textural attributes, the same tendencies were observed for both lentil products, i.e., softer cookies were obtained upon xylitol and WP addition, while hardness increased upon inulin enrichment. Stickiness was differently influenced by the functional ingredients in the case of green and red lentil cookies, but all the xylitol-containing cookies were less crumbly than the controls.

Sensory attributes changed upon sugar replacement by xylitol in terms of softer, less crunchy, less sweet, and less homogeneous products. When inulin and WP were used in combination, the properties were significantly different from those of the other enriched or sugar-free cookies for both lentil types (surface homogeneity and color, sweet taste, hardness, crunchiness, and crumbliness).

Our findings demonstrated that different enrichments considerably affected many technological, textural, and sensory properties of both the green or red lentil cookies in different ways, particularly regarding the impact of inulin and xylitol on the green lentil cookies. The investigation of the roles and interactions between the different functional ingredients uncovered their functions in gluten-free lentil-based cookies. These results may serve as orientation points in the further development of functional GF products.

## Figures and Tables

**Figure 1 foods-11-03819-f001:**
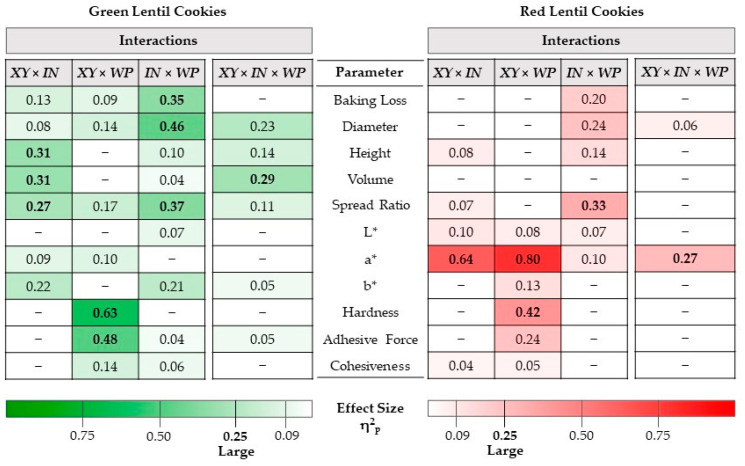
Results of factorial ANOVA analysis performed on lentil cookies’ data. The partial eta-squared (η^2^_p_) of the interactions is shown only for significant values (*p* < 0.05). Mark ‘−’ means that the interaction effect was not significant (*p* > 0.05). XY: xylitol, IN: inulin, WP: whey protein isolate.

**Table 1 foods-11-03819-t001:** Recipe of the prepared cookies.

Ingredient	% ^a^	Amount (g)							
		Control	XY	IN	WP	XY-IN	XY-WP	IN-WP	XY-IN-WP
Lentil flour (green or red)	29.7–47.6	100.0 (47.6%)	100.0 (47.6%)	87.4 (41.6%)	75.0 (35.7%)	87.4 (41.6%)	75.0 (35.7%)	62.4 (29.7%)	62.4 (29.7%)
Powdered sugar	27.5 or 0.0	57.8	-	57.8	57.8	-	-	57.8	-
Glucose solution (5 g/100 mL)	7.0 or 0.0	14.6	-	14.6	14.6	-	-	14.6	-
Distilled water	3.4 or 10.0	7.1	21.0	7.1	7.1	21.0	21.0	7.1	21.0
Margarine (with 70% fat content)	13.5	28.4	28.4	28.4	28.4	28.4	28.4	28.4	28.4
Sodium bicarbonate	0.5	1.11	1.11	1.11	1.11	1.11	1.11	1.11	1.11
Salt	0.4	0.93	0.93	0.93	0.93	0.93	0.93	0.93	0.93
Xylitol	0.0 or 27.9	-	58.5	-	-	58.5	58.5	-	58.5
Inulin	0.0 or 6.0	-	-	12.6	-	12.6	-	12.6	12.6
Whey protein isolate	0.0 or 11.9	-	-	-	25.0	-	25.0	25.0	25.0

^a^ Proportion of ingredients in total formulation where the sum of all ingredients is 210 g. Mark ‘-‘ means that the ingredient was not included in the recipe.

**Table 2 foods-11-03819-t002:** Baking loss and geometry of green lentil cookies.

Sample	Baking Loss (%)	Diameter (mm)	Height (mm)	Volume (cm^3^)	Spread Ratio (-)
GL (control)	17.0 ± 0.4 bc	61.5 ± 3.1 d	7.1 ± 0.3 cd	21.2 ± 2.4 d	8.6 ± 0.6 cd
GL-XY	16.9 ± 0.9 abc	48.7 ± 0.4 a	5.9 ± 0.4 ab	10.9 ± 0.8 a	8.4 ± 0.6 bcd
GL-IN	18.2 ± 0.3 d	64.3 ± 2.2 e	5.1 ± 0.6 a	16.5 ± 2.3 b	12.8 ± 1.7 e
GL-WP	16.9 ± 0.4 abc	62.8 ± 1.0 de	8.3 ± 0.4 e	25.8 ± 1.3 e	7.6 ± 0.4 ab
GL-XY-IN	17.4 ± 0.4 c	58.5 ± 1.9 c	6.5 ± 0.4 bc	17.4 ± 1.5 bc	9.1 ± 0.7 d
GL-XY-WP	17.5 ± 0.7 c	58.0 ± 2.1 bc	7.7 ± 1.2 de	20.4 ± 3.6 d	7.7 ± 1.3 abc
GL-IN-WP	16.6 ± 0.4 ab	62.4 ± 1.5 de	8.1 ± 0.5 e	24.8 ± 1.8 e	7.7 ± 0.6 abc
GL-XY-IN-WP	16.3 ± 0.8 a	55.5 ± 1.9 b	8.2 ± 0.9 e	19.7 ± 2.0 cd	6.9 ± 1.0 a

GL: green lentil flour, XY: xylitol, IN: inulin, WP: whey protein isolate. Data presented as the mean ± the standard deviation (*n* = 12). Within each column, different letters represent the significant difference between means (*p* < 0.05; factorial ANOVA, Tukey’s HSD test).

**Table 3 foods-11-03819-t003:** Baking loss and geometry of red lentil cookies.

Sample	Baking Loss (%)	Diameter (mm)	Height (mm)	Volume (cm^3^)	Spread Ratio (-)
RL (control)	16.9 ± 0.9 a	63.2 ± 3.2 bc	6.9 ± 0.5 bc	21.8 ± 2.4 a	9.2 ± 0.8 c
RL-XY	17.3 ± 0.3 abc	58.0 ± 1.6 a	8.0 ± 0.4 de	21.1 ± 1.5 a	7.3 ± 0.4 a
RL-IN	17.4 ± 0.4 bc	67.4 ± 4.9 d	6.1 ± 0.4 a	21.9 ± 3.7 a	11.1 ± 1.1 d
RL-WP	17.8 ± 0.2 c	64.8 ± 1.4 bcd	7.6 ± 0.4 d	25.0 ± 1.4 b	8.6 ± 0.5 bc
RL-XY-IN	17.9 ± 0.2 c	66.8 ± 1.8 d	6.4 ± 0.5 ab	22.4 ± 2.5 ab	10.5 ± 0.9 d
RL-XY-WP	17.8 ± 0.2 c	61.6 ± 3.3 b	8.4 ± 0.7 e	25.0 ± 2.3 b	7.4 ± 1.0 a
RL-IN-WP	17.0 ± 0.2 ab	65.6 ± 1.0 cd	7.5 ± 0.3 cd	25.3 ± 1.0 b	8.8 ± 0.4 bc
RL-XY-IN-WP	17.7 ± 0.6 c	61.3 ± 2.6 ab	7.8 ± 0.8 de	23.0 ± 2.6 ab	8.0 ± 1.0 ab

RL: red lentil flour, XY: xylitol, IN: inulin, WP: whey protein isolate. Data presented as the mean ± the standard deviation (*n* = 12). Within each column, different letters represent the significant difference between means (*p* < 0.05; factorial ANOVA, Tukey’s HSD test).

**Table 4 foods-11-03819-t004:** Color parameters of green lentil cookies.

Sample	L*	a*	b*	∆E
GL (control)	35.68 ± 6.80 bcd	6.82 ± 0.86 c	16.24 ± 1.82 d	-
GL-XY	32.89 ± 0.72 ab	3.45 ± 0.33 a	8.15 ± 0.99 a	9.2
GL-IN	34.46 ± 1.28 abc	6.65 ± 0.49 c	13.13 ± 1.08 bc	3.3
GL-WP	37.77 ± 1.13 cd	9.97 ± 0.30 d	18.50 ± 1.10 e	4.4
GL-XY-IN	31.17 ± 0.88 a	4.34 ± 0.40 b	8.77 ± 0.95 a	9.1
GL-XY-WP	32.56 ± 1.04 ab	6.29 ± 0.24 c	12.26 ± 1.00 b	5.1
GL-IN-WP	39.17 ± 1.25 d	10.03 ± 0.40 d	19.08 ± 0.90 e	5.5
GL-XY-IN-WP	35.58 ± 1.14 bcd	6.60 ± 0.57 c	14.31 ± 1.01 c	1.9

GL: green lentil flour, XY: xylitol, IN: inulin, WP: whey protein isolate. Data presented as the mean ± the standard deviation (*n* = 12). Within each column, different letters represent the significant difference between means (*p* < 0.05; factorial ANOVA, Tukey’s HSD test). Delta E (∆E) refers to the total color difference compared to the control.

**Table 5 foods-11-03819-t005:** Color parameters of red lentil cookies.

Sample	L*	a*	b*	∆E
RL (control)	45.25 ± 2.74 a	16.53 ± 0.73 e	28.17 ± 3.04 a	-
RL-XY	53.01 ± 1.41 c	7.61 ± 0.33 a	32.29 ± 1.08 b	12.5
RL-IN	43.86 ± 1.72 a	15.31 ± 1.11 d	26.09 ± 2.93 a	2.8
RL-WP	43.22 ± 1.48 a	16.82 ± 0.52 e	26.40 ± 1.73 a	2.7
RL-XY-IN	48.05 ± 2.22 b	10.93 ± 0.29 b	32.21 ± 5.69 b	7.5
RL-XY-WP	52.64 ± 1.54 c	14.01 ± 0.35 c	36.17 ± 1.25 c	11.2
RL-IN-WP	43.29 ± 1.53 a	17.78 ± 0.31 f	26.76 ± 1.66 a	2.7
RL-XY-IN-WP	50.75 ± 2.90 bc	16.67 ± 0.37 e	36.02 ± 2.38 c	9.6

RL: red lentil flour, XY: xylitol, IN: inulin, WP: whey protein isolate. Data presented as the mean ± the standard deviation (*n* = 12). Within each column, different letters represent the significant difference between means (*p* < 0.05; factorial ANOVA, Tukey’s HSD test). Delta E (∆E) refers to the total color difference compared to the control.

**Table 6 foods-11-03819-t006:** Textural properties of green lentil cookies.

Sample	Hardness (g)	Adhesive Force (g)	Cohesiveness (-)
GL (control)	612.0 ± 152.5 c	39.4 ± 28.0 ab	0.04 ± 0.01 a
GL-XY	135.0 ± 24.6 a	46.3 ± 6.4 ab	0.19 ± 0.03 c
GL-IN	820.9 ± 190.5 d	59.5 ± 36.0 b	0.06 ± 0.02 a
GL-WP	350.1 ± 65.7 b	12.3 ± 8.8 a	0.03 ± 0.01 a
GL-XY-IN	290.8 ± 68.5 ab	60.6 ± 15.4 b	0.21 ± 0.06 c
GL-XY-WP	446.5 ± 103.0 b	106.7 ± 35.8 c	0.15 ± 0.04 b
GL-IN-WP	435.8 ± 102.8 b	29.0 ± 19.8 ab	0.03 ± 0.01 a
GL-XY-IN-WP	639.4 ± 178.8 c	174.3 ± 58.5 d	0.13 ± 0.01 b

GL: green lentil flour, XY: xylitol, IN: inulin, WP: whey protein isolate. Data presented as the mean ± the standard deviation (*n* = 12). Within each column, different letters represent the significant difference between means (*p* < 0.05; factorial ANOVA, Tukey’s HSD test).

**Table 7 foods-11-03819-t007:** Textural properties of red lentil cookies.

Sample	Hardness (g)	Adhesive Force (g)	Cohesiveness (-)
RL (control)	762.1 ± 218.5 d	98.5 ± 82.2 bc	0.05 ± 0.02 a
RL-XY	177.3 ± 46.9 a	38.4 ± 9.8 a	0.23 ± 0.04 b
RL-IN	964.0 ± 206.0 e	127.0 ± 83.9 c	0.05 ± 0.02 a
RL-WP	397.7 ± 145.5 bc	23.4 ± 24.2 a	0.03 ± 0.02 a
RL-XY-IN	228.4 ± 58.6 ab	48.0 ± 15.7 ab	0.28 ± 0.04 c
RL-XY-WP	252.8 ±73.8 ab	61.9 ± 23.0 ab	0.19 ± 0.07 b
RL-IN-WP	467.3 ± 126.6 c	25.9 ± 21.2 a	0.04 ± 0.02 a
RL-XY-IN-WP	274.8 ± 44.2 ab	75.9 ± 16.3 abc	0.21 ± 0.06 b

RL: red lentil flour, XY: xylitol, IN: inulin, WP: whey protein isolate. Data presented as the mean ± the standard deviation (*n* = 12). Within each column, different letters represent the significant difference between means (*p* < 0.05; factorial ANOVA, Tukey’s HSD test).

**Table 8 foods-11-03819-t008:** Sensory properties of green lentil cookies.

	GL (Control)	GL-XY-IN	GL-XY-WP	GL-IN-WP	GL-XY-IN-WP
Surface homogeneity	70 A	44 ± 19 b,B	18 ± 15 a,B	85 ± 11 c,B	32 ± 16 ab,B
Surface color	40 A	26 ± 23 a,A	22 ± 15 a,B	82 ± 12 b,B	33 ± 17 a,A
Baked aroma	10 A	38 ± 28 a,B	22 ± 25 a,A	33 ± 26 a,B	34 ± 25 a,B
Lentil aroma	30 A	41 ± 28 b,A	14 ± 23 a,B	14 ± 13 a,B	16 ± 20 a,B
Lentil taste	40 A	56 ± 24 a,B	36 ± 25 a,A	33 ± 28 a,A	35 ± 21 a,A
Sweet taste	60 A	38 ± 27 a,B	37 ± 18 a,B	62 ± 25 b,A	27 ± 13 a,B
Hardness	70 A	15 ± 14 a,B	32 ± 13 b,B	76 ± 17 c,A	28 ± 14 ab,B
Crunchiness	80 A	28 ± 17 a,B	40 ± 15 a,B	83 ± 16 b,A	37 ± 16 a,B
Crumbliness	30 A	51 ± 30 a,B	55 ± 27 a,B	32 ± 19 a,A	51 ± 31 a,B

GL: green lentil flour, XY: xylitol, IN: inulin, WP: whey protein isolate. Data presented as the mean ± the standard deviation (*n* = 13). Within each row, different lower-case letters represent the significant difference between means (*p* < 0.05; one-way ANOVA, Tukey’s HSD test). Within each row, different upper-case letters represent the significant difference compared to the control value (*p* < 0.05; one-sample Student’s *t*-test).

**Table 9 foods-11-03819-t009:** Sensory properties of red lentil cookies.

	RL (Control)	RL-XY-IN	RL-XY-WP	RL-IN-WP	RL-XY-IN-WP
Surface homogeneity	70 A	46 ± 16 a,B	45 ± 23 a,B	84 ± 09 b,B	42 ± 15 a,B
Surface color	70 A	49 ± 21 b,B	28 ± 16 a,B	78 ± 14 c,B	27 ± 17 a,B
Baked aroma	80 A	66 ± 23 b,B	34 ± 27 a,B	57 ± 29 ab,B	41 ± 28 ab,B
Lentil aroma	50 A	68 ± 26 b,B	42 ± 27 a,A	40 ± 25 a,A	36 ± 25 a,B
Lentil taste	40 A	68 ± 22 b,B	43 ± 31 ab,A	28 ± 23 a,B	41 ± 29 a,A
Sweet taste	60 A	44 ± 26 a,B	40 ± 29 a,B	70 ± 18 b,B	37 ± 25 a,B
Hardness	70 A	17 ± 14 a,B	36 ± 15 b,B	78 ± 16 c,A	33 ± 13 b,B
Crunchiness	80 A	26 ± 13 a,B	44 ± 16 b,B	87 ± 10 c,B	42 ± 17 b,B
Crumbliness	30 A	76 ± 17 b,B	62 ± 15 b,B	22 ± 16 a,A	68 ± 19 b,B

RL: red lentil flour, XY: xylitol, IN: inulin, WP: whey protein isolate. Data presented as the mean ± the standard deviation (*n* = 16). Within each row, different lower-case letters represent the significant difference between means (*p* < 0.05; one-way ANOVA, Tukey’s HSD test). Within each row, different upper-case letters represent the significant difference compared to the control value (*p* < 0.05; one-sample Student’s *t*-test).

## Data Availability

The data presented in this study are available upon request from the corresponding author.

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
