# Peer review of "Development of Protein- and Fiber-Enriched, Sugar-Free Lentil Cookies: Impact of Whey Protein, Inulin, and Xylitol on Physical, Textural, and Sensory Characteristics"

_foods, 2022, doi:10.3390/foods11233819_

Round 1
Reviewer 1 Report
I am very grateful you for the invitation to review the manuscript foods-2050479 by Hajas and coauthors "The production of protein- and fiber-enriched, sugar-free lentil cookies: Physical, textural and sensory characteristics”. The aim of this research was to enhance the nutritionally favorable properties of lentil-based cookies in terms of protein and fiber content, as well as to develop a sugar-free product. Furthermore, the study also aimed to increase understanding of the functionality of the different functional ingredients. The work is interesting but needs adjustments to increase the quality of the material.
Comments:
- The title must be modified since nutritional values and the increase in protein and fiber are not presented. Only the technological aspects are taken into consideration.
- Abstract: Please introduce the gluten issue before demonstrating the diet issues.
- Abstract, Line 11: Insert the concentrations evaluated.
- Abstract, Lines 13-14: The sentence can be removed, without prejudice to the presentation of the work.
- Abstract: Insert the main (numerical) results in the abstract.
- Abstract: Insert the conclusion in the abstract.
- Line 24: Change the repeated keywords by different words from the title.
- Introduction: Please enter the updated numbers of people affected by celiac disease and the consequences caused by the disease.
- Lines 44-45: The authors presented a compound yearly growth rate, but the overall market value was not presented. Please enter global data.
- Introduction: Please highlight the problem of incorporating proteins and fibers in bakery products (structure, sensory, among others).
- Lines 54-55: Specify if there are any restrictions that lead to this.
- Line 60: Please check the information: harder is different from “better hardness”.
- Lines 65-71: Rewrite the sentence in the past tense and more simply.
- 2.1. Materials: The item is intended for the presentation of the materials used. The preparation methodology must be presented in a separate item.
- Lines 80-81: Authors should briefly describe the cookie production methodology in a separate item.
- Lines 80-94 and Table 1: Please also enter the specific percentage of each component.
- Line 94: Enter the rest time between the cooking process and the analysis.
- Line 175: The sentence is so generic and doesn't add information. Authors should review and complement.
- Line 233-234: Indicate whether in gluten-free products and without the presence of protein structure the same would occur.
- Line 340-342: Please insert the reference.
- Line 451: Please check the proper citation format.
- Lines 8; 9; 65: Despite the importance of nutritional increment in cookies being highlighted, no data is presented throughout the work. Please enter nutritional data or change sentences.
Reviewer 2 Report
This manuscript investigated the potential of green and red lentil-based gluten-free cookies enriched with different combinations of whey protein, dietary fiber (inulin) and xylitol as sweetener. Suggestions for modification are as follows:
Line 11, Why choose xylitol as sweetener?
Line 75, What is the size of the sieve?
Line 179, Why did inulin enrichment led to significantly higher baking loss?
Line 204-205, Why did addition of WP increase the diameter significantly only in xylitol- containing cookies?
Line 216-217, Why did WP induce an increase in height in xylitol-containing samples?
Line 247-248, Why did inulin produce a significant increase of spreading, while WP induced a significantly lower ratio?
Line 255-256, Why did the presence of xylitol resulted in no significant change of baking losses?
Line 262-263, Why did the presence of xylitol led to a decrease in diameter and increase in the presence of inulin?
Line 271-272, Why did inulin produced a decrease in cookie height?
Line 279-280, Why did inulin produce an increase, while WP lead to a decrease of the ratio? Please cite references to prove your point.
Line 289, When the author analyze the interactions occurring when different enrichments are applied, please quote relevant references.
Line 348-349, Could the author explain why addition of xylitol, WP or inulin did not induce any significant changes in L* compared to the control?
Line 397-399, “it can be concluded that in terms of all color attributes replacement of sugar by xylitol had the highest impact”. Please cite relevant references to prove your point.
Line 408-409, Could the author explain why Cookies containing xylitol were softer?
Line 410, Could the author explain why Cookies containing xylitol were softer?
Line 436-439, Please cite relevant references to prove your results of hardness.
Line 472-473, Could the author explain why sensory profile analysis of red lentil cookies showed significant differences for all the properties tested? Please cite relevant references to prove your results.
Tables:
Modify the variance in Table2-Table9
Reviewer 3 Report
The present manuscript has investigated the influence of inulin, whey protein, and xylitol on the functional properties of lentil based gluten-free cookies. However, the effects of these ingredients have been investigated by several researchers and the novelty of this manuscript is not highlighted.
- Many of the results are attributed to the chemical composition of cookies but it is not reported in the manuscript.
- The results are just reported but they are not discussed properly.
other comments:
Line 80: write a section “preparation of lentil-based cookies” in methods and explain the process
- Why you have used TPA test for texture measurement of cookies? (Triple point bend is more common)
Round 2
Reviewer 1 Report
After carefully checking the revisions, the manuscript is suggested to be accepted by Foods;
Reviewer 3 Report
The manuscript is acceptable.